# Substantial underestimation of SARS-CoV-2 infection in the United States

Sean L. Wu [1], Andrew N. Mertens [1], Yoshika S. Crider [1,2], Anna Nguyen [1], Nolan N. Pokpongkiat [1], Stephanie Djajadi [1], Anmol Seth[1], Michelle S. Hsiang [3,4,5], John M. Colford Jr.[1], Art Reingold [1], Benjamin F. Arnold[6,7], Alan Hubbard[1] & Jade Benjamin-Chung [1]✉

Accurate estimates of the burden of SARS-CoV-2 infection are critical to informing pandemic response. Confirmed COVID-19 case counts in the U.S. do not capture the total burden of the pandemic because testing has been primarily restricted to individuals with moderate to severe symptoms due to limited test availability. Here, we use a semi-Bayesian probabilistic bias analysis to account for incomplete testing and imperfect diagnostic accuracy. We estimate 6,454,951 cumulative infections compared to 721,245 confirmed cases (1.9% vs. 0.2% of the population) in the United States as of April 18, 2020. Accounting for uncertainty, the number of infections during this period was 3 to 20 times higher than the number of confirmed cases. 86% (simulation interval: 64–99%) of this difference is due to incomplete testing, while 14% (0.3–36%) is due to imperfect test accuracy. The approach can readily be applied in future studies in other locations or at finer spatial scale to correct for biased testing and imperfect diagnostic accuracy to provide a more realistic assessment of COVID-19 burden.

[1] Division of Epidemiology and Biostatistics, University of California, 2121 Berkeley Way, Berkeley, CA 94720-7360, USA. [2] Energy and Resources Group, University of California, 310 Barrows Hall, Berkeley, CA 94720-3050, USA. [3] Department of Pediatrics, University of Texas Southwestern Medical Center, 5323 Harry Hines Blvd, Dallas, TX 75390-9003, USA. [4] Pandemic Community Response and Resilience Initiative, Global Health Group, University of California, San Francisco, Mission Hall, Box 1224, 550 16th Street, Third Floor, San Francisco, CA 94158, USA. [5] Department of Pediatrics, University of California, San Francisco 550 16th StreetBox 0110San Francisco, CA 94143, USA. [6] Francis I. Proctor Foundation, University of California, San Francisco 95 Kirkham Street, San Francisco, CA 94143, USA. [7] Department of Ophthalmology, University of California, San Francisco 10 Koret Way, San Francisco, CA 94143-0730, USA. ✉email: jadebc@berkeley.edu

The severe acute respiratory syndrome-coronavirus 2 (SARS-CoV-2) pandemic is reported to have caused 2,003,930 confirmed cases of coronavirus disease 2019 (COVID-19) in the U.S. by June 11, 2020. The first known case in the U.S. was confirmed on January 21, 2020. In February, SARS-CoV-2 testing remained limited due to flawed test kits. For the first few months of the pandemic, the U.S. Centers for Disease Control and Prevention (CDC) recommended that physicians prioritize testing hospitalized patients, who tend to have moderate to severe symptoms. Most state testing policies were consistent with this recommendation (Supplementary Table 1). Yet, evidence from studies that conducted broader testing suggest that 30–70% of individuals who test positive have mild or no symptoms[1–4] and that asymptomatic and pre-symptomatic individuals can transmit SARS-CoV-2[5–7]. Thus, a substantial number of mild or asymptomatic infections in the U.S. may be undetected[8]. Furthermore, initial evidence suggests that tests based on nasopharyngeal and throat swabs may produce false negative results[9–12]. Thus, counts of confirmed cases are biased due to incomplete testing and imperfect test sensitivity. Accurate estimates of the burden of SARS-CoV-2 infection are critical to understanding the course of the pandemic and informing public health response[8,13]. Furthermore, limited and biased testing can influence estimates of SARS-CoV-2 transmissibility, which typically rely upon observed counts of cumulative infections[14].

To date, the majority of studies that have estimated the burden of SARS-CoV-2 infection have used mathematical models (e.g., compartmental or agent-based models)[15–19]. Mathematical modeling studies attempt to mimic natural disease transmission systems by modeling age and social structure, travel and commuting patterns, and immune dynamics. Such models are particularly useful for projecting transmission patterns under hypothetical interventions but can be highly complex; when sufficient data do not exist to parameterize such models, results can be quite biased[20,21]. In addition, model outputs are sensitive to assumed population structure and contact patterns, which are difficult to validate, particularly in a novel pathogen setting.

We estimate the total number of SARS-CoV-2 infections in each U.S. state from February 28 to April 18, 2020 using probabilistic bias analysis, a semi-Bayesian approach, to correct empirical confirmed case counts for bias due to incomplete testing and imperfect test accuracy. This method is commonly used to quantify the impact of and correct for measurement bias in observational epidemiologic studies[22]. Our objective is to estimate infection burden from empirical data rather than to forecast future dynamics. In the absence of systematic random sampling or robust surveillance, true COVID-19 incidence is unknown and "we are operating in the dark"[23]. This being the case, our method provides an estimate of the true number of infections, which can help not only determine what kind of response is appropriate, but also evaluate the progress or failure of mitigation or containment efforts. Further, the simplicity of our approach facilitates transparent assessment of modeling assumptions.

## Results

**SARS-CoV-2 testing rates varied widely by state**. We quantified the SARS-CoV-2 testing rate per 1000 persons using daily counts of tests in each state and 2019 projected state populations from the 2010 U.S. Census[24]. By April 18, 2020, the SARS-CoV-2 testing rate was 11 per 1000 in the U.S. However, there were large discrepancies in testing between states, with state-level testing rates of 6 per 1000 in Kansas to 31 per 1000 in Rhode Island (Fig. 1). Generally, testing rates were higher in the Northwest and Northeast and lower in the Midwest and South.

**Confirmed COVID-19 case counts underestimate infections**. When correcting for incomplete testing and imperfect test accuracy, we estimated that the total number of SARS-CoV-2 infections in the U.S. by April 18, 2020 was 6,454,951 (19 per 1000)—an estimate nine times larger than the 721,245 confirmed cases (2 per 1000) reported during this period. These results imply that 89% of infections in the U.S. were undocumented. This finding is consistent with a mathematical modeling study that reported that 86% of infections were undocumented using data from Wuhan, China[18]. The 95% simulation interval for the number of estimated infections in the U.S. was 2,240,740–14,856,756 (Supplementary Fig. 1). This corresponds to an estimated number of SARS-CoV-2 infections 3 to 20 times higher than the number of confirmed cases. Nationally, we estimate that 84% (simulation interval: 64–99%) of the difference between confirmed cases and estimated infections was due to incomplete testing, and 16% (simulation interval: 0.3–36%) of the difference was due to imperfect test accuracy.

Disparities between confirmed SARS-CoV-2 infections and estimated total infections varied widely by state and geographic region. In each state, confirmed COVID-19 case counts ranged from 0.4 to 12.2 per 1000, while estimated total infections ranged from 3.1 to 65.0 per 1000 (Fig. 2a, Supplementary Fig. 1). Compared to confirmed COVID-19 case counts, expected infections were 5 to 33 times larger (Fig. 2b). COVID-19 incidence was highest in the northeast, Midwest, and Louisiana using confirmed case counts (Fig. 3a) or estimated infections (Fig. 2a). However, underestimation of SARS-CoV-2 infections was more common in Puerto Rico, California, the Midwest, and some Southern states (Fig. 3b). In 33 states, the number of infections was at least 10 times larger than the number of confirmed cases. Differences in state-specific results are driven by observed differences in transmission, testing rates, and test positivity rates in each state rather than our modeling assumptions. In states with the largest underestimates of SARS-CoV-2 infections, public health responses based on confirmed cases may be inadequate to reduce transmission.

**Sensitivity analyses produced similar estimates**. Our approach and mathematical models both rely on constraints and assumptions based on available empirical data. However, available data are nonetheless limited. To demonstrate the robustness of our results to alternative assumptions, we conducted sensitivity analyses under different plausible scenarios. Sensitivity analyses used alternative prior distributions for the parameters that had the least available published evidence and allowed for correlation between two of our priors that our model assumed were independent (Supplementary Table 2). For all scenarios but one, results were robust to the changes in prior distributions. In the scenario that increased the upper bound and mean of the distribution of the probability of testing positive among untested individuals with mild or no symptoms, estimates of total cumulative infections were higher, particularly in states with higher test positivity (Supplementary Fig. 2). Overall, the consistency of our findings under different prior distributions supports the robustness of our approach.

## Discussion

Our findings illustrate the importance of adjusting estimates of COVID-19 infections for testing practices and diagnostic accuracy during a period of low testing rates. A strength of our approach is that it quantifies the contribution of incomplete testing vs. imperfect test accuracy to underestimation of the burden of COVID-19, demonstrating that the majority was due to incomplete testing. Our methods are not specific to the diagnostic

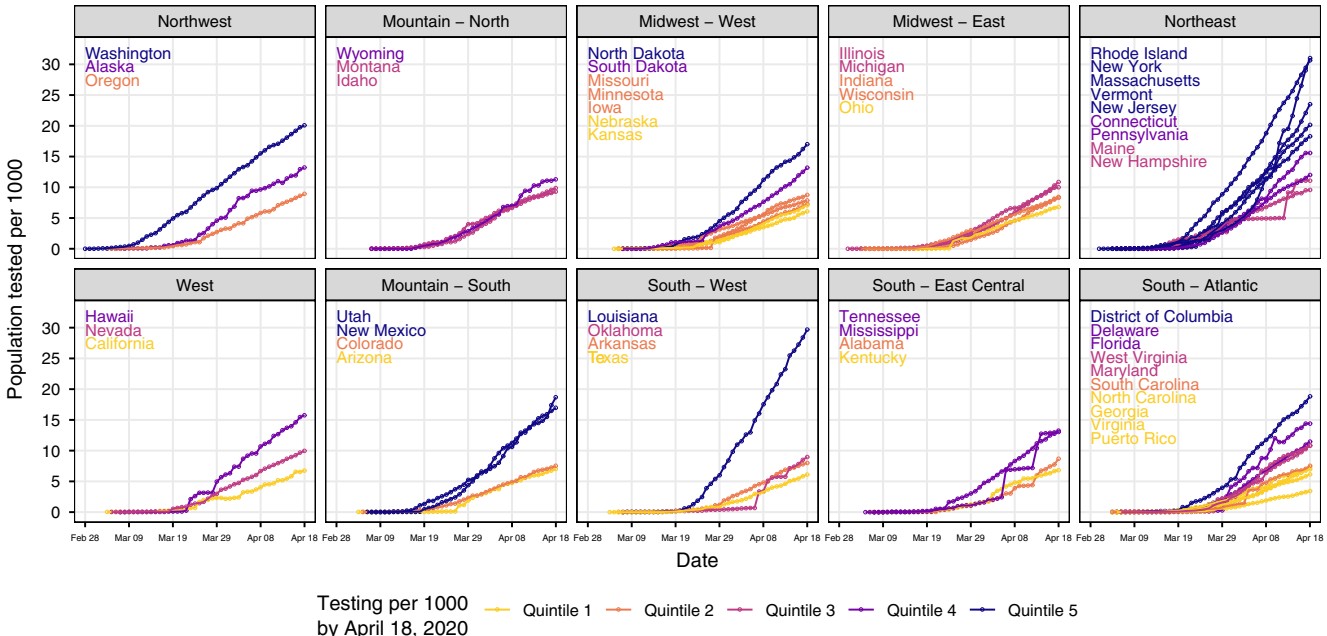

**Fig. 1 Daily SARS-CoV-2 tests per 1000 population in the United States by state and region.** SARS-CoV-2 testing rates in US states increased from close to 0 in early March to 6 per 1000 in Kansas to 31 per 1000 in Rhode Island by April 18, 2020. Generally, testing rates were higher in the Northwest and Northeast and lower in the Midwest and South. We estimated the cumulative population tested in each state by date by dividing the number of tests performed by 2019 population projections from the U.S. 2010 Census. Each open circle indicates the number of individuals tested in a state on a given day per 1000. Line, point, and text colors are based on quintiles of the distribution of testing per 1000 population on April 18, 2020 across all states. Testing quintile from lowest to highest is mapped to color from warmest to coolest (such that quintile 1, the lowest, maps to yellow (warmest), and quintile 5, the highest, maps to dark purple (coolest)). In each panel, state names are sorted in descending order by the population tested per 1000. Quality of daily estimates of the number of tests performed varied by state; see Supplementary Table 3 for details. See interactive plot at https://covid19epi.github.io/stats/.

test used and can be applied, for example, to serological studies to account for the imperfect diagnostic accuracy of serological tests and non-random sampling strategies used in some serological studies.

Little is known about whether infection with SARS-CoV-2 confers lasting immunity, and if so, for how long. Even in a best-case scenario in which SARS-CoV-2 infection produces immunity for 1–2 years, as is common for other betacoronaviruses[25], our results contribute to growing consensus that a very small proportion of the population has developed immunity and that the U.S. is not close to achieving herd immunity[26,27].

Our findings are broadly consistent with other studies using different methods to estimate the total SARS-CoV-2 infections. A mathematical modeling study projected 560 cumulative infections per 1000 in the U.S. in 2020 if social distancing had been implemented population-wide[16]. In comparison, we estimated 19 infections per 1000 over a 25-day period in an early stage of the pandemic prior to peak transmission. Another study using a metapopulation transmission model estimated that there were between 10,000 and 30,000 new cases per day during our study period, which translates to similar but lower infection counts[28]. A study comparing influenza-like illness data in 2020 to prior years in the U.S. reported similar rankings of the cumulative incidence of SARS-CoV-2 infections by state[29]. Other studies from countries where time-indexed district-level hospitalization and mortality data are readily available estimated the proportion of the population infected with SARS-CoV-2 in Italy and France, where testing rates are higher[30,31]. These studies estimate that the number of infections was at least twice the number of confirmed cases, consistent with our findings.

At least four studies conducted in the U.S. within the same time frame as this study have reported SARS-CoV-2 seroprevalence ranging from 0.4% in Seattle, Washington to 4.4% in Baton Rouge, Louisiana[32–34]. Here, we estimated that the cumulative incidence in the U.S. was close to 2% by mid-April, with wide variation between states. However, specificity of serological assays was poor in at least two of these studies, so estimates may not be reliable, particularly given how the low seroprevalence observed. Furthermore, given that little is known about the quantity and duration of SARS-CoV-2 antibodies following infection, comparisons of seroprevalence estimates with cumulative incidence estimates based on current infection in this study must be made cautiously[35].

As our method does not incorporate a mechanistic transmission model, we are unable to make quantitative statements about future dynamics (i.e., forecasts). Rather, our method provides a more realistic picture of infection burden at a given point in time, adjusting for biases induced by differential testing practices and characteristics. In addition, our prior distributions are based on limited initial evidence about SARS-CoV-2 testing probabilities. Though we used the best available evidence at this early stage of the pandemic, it is possible that our priors do not reflect true testing probabilities. For example, our testing priors were informed by state-level testing guidelines, which typically prioritize groups for testing and allow physician discretion in ordering tests. In some areas, changes in testing capacity and protocols over time may imply different priors in these states (Supplementary Table 1); however, test supply shortages continue to be reported in the media, and state-level guidance to emphasize testing individuals thought to be at greatest risk.

Another limitation is that our model used state-specific estimates of the probability of testing positive among individuals who were tested. In states with very low testing rates, empirical test positive probabilities may not accurately reflect incidence in the

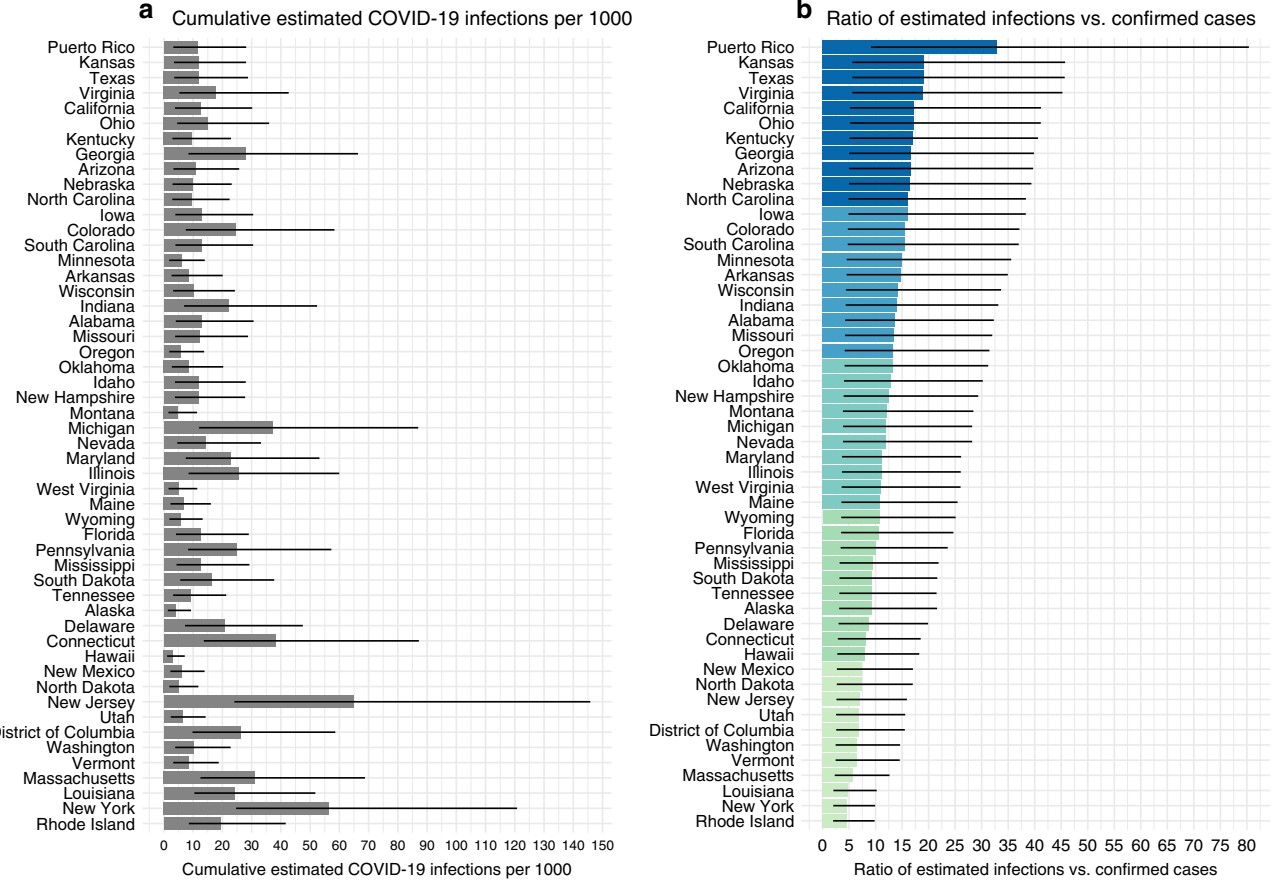

**Fig. 2 Confirmed COVID-19 cases vs. estimated SARS-CoV-2 infections.** In each US state, confirmed COVID-19 case counts ranged from 0.4 to 12.2 per 1000, while estimated total infections ranged from 3.0 to 63.0 per 1000. **a** Gray bars indicate the median of sampled distribution of estimated infections from probabilistic bias analysis. **b** Ratio between estimated infections versus confirmed cases, with reveals underestimation of total SARS-CoV-2 infection burden according to our model. In (**b**), ratios in each state are colored by quintile in descending order, with the darkest shade of blue indicating the largest quintile, and the lightest shade of green indicating the lowest quintile. Analyses include cumulative confirmed COVID-19 case counts up to April 18, 2020. Estimated SARS-CoV-2 infections were from a Bayesian probabilistic bias analysis to correct for incomplete testing and imperfect test accuracy. Estimated infections include both symptomatic and asymptomatic infections. Horizontal black lines indicate the simulation interval for estimated infections (2.5th and 97.5th percentiles of the distribution of estimated infections for each state) which were computed via $10^4$ Monte Carlo samples from the distribution of estimated SARS-CoV-2 infections in each state. Quality of daily estimates of the number of tests performed varied by state; see Supplementary Table 3 for details. See interactive plot at https://covid19epi.github.io/stats/.

general population due to prioritized testing of severely ill and special populations. Nevertheless, our wide simulation intervals reflect these uncertainties in our prior distributions. Even using the lower bound of the simulation interval for the U.S. as a conservative estimate suggests that confirmed case counts underestimate total infections by a factor of 3. Results from future studies that rigorously estimate the incidence of symptomatic and asymptomatic SARS-CoV-2 infection can be used to update our priors and improve the precision and accuracy of our estimates.

State-level data on SARS-CoV-2 positive tests may mask meaningful geographic variation at a smaller scale. Unfortunately, county-level estimates of the number of individuals tested were not readily available at the time of this study. In addition, our model did not account for state-specific variation in data quality due to incomplete reporting of testing in some states (Supplementary Table 3).

We did not estimate the infection fatality ratio (IFR) because COVID-19 deaths outside hospitals are likely to be under-reported, and death registrations may not be up to date. COVID-19-specific death reporting rates likely differ between tested and untested individuals and individuals with different co-morbidities, but the extent of this difference is difficult to quantify. In

addition, accurate estimation of the IFR would account for age, but age-stratified counts of COVID-19 deaths in each state are not readily available. As such, it is difficult to parameterize a plausible method to correct IFRs for bias[8]. One recent study employed a fully probabilistic Bayesian model to estimate IFRs[36], but its data and priors are subject to the same limitations as this study; as a result, its model was only partially identifiable, and the mean IFR could only be estimated within an interval.

Underestimates of the number of SARS-CoV-2 infections jeopardize the success of pandemic response policies: they suggest to the public that the threat of the pandemic is smaller than it is, making it difficult to justify stringent social distancing policies. Our results highlight the urgent need to systematically expand SARS-CoV-2 testing across the U.S. to provide an accurate evidence base for pandemic response policies.

## Methods
**Data.** We obtained 2019 projected state populations ($N$) from the 2010 U.S. Census and observed daily counts of tests ($N_{tested}$) and confirmed SARS-CoV-2 positive tests ($N_{tested}^+$) in each state from February 28 to April 18, 2020. Data were collected by the COVID Tracking Project, which assembles data on a regular basis primarily from state, district, and territory public health departments. National case counts from this source were comparable with those from the CDC on April 18 (COVID

**a**  Cumulative confirmed COVID-19 cases per 1000

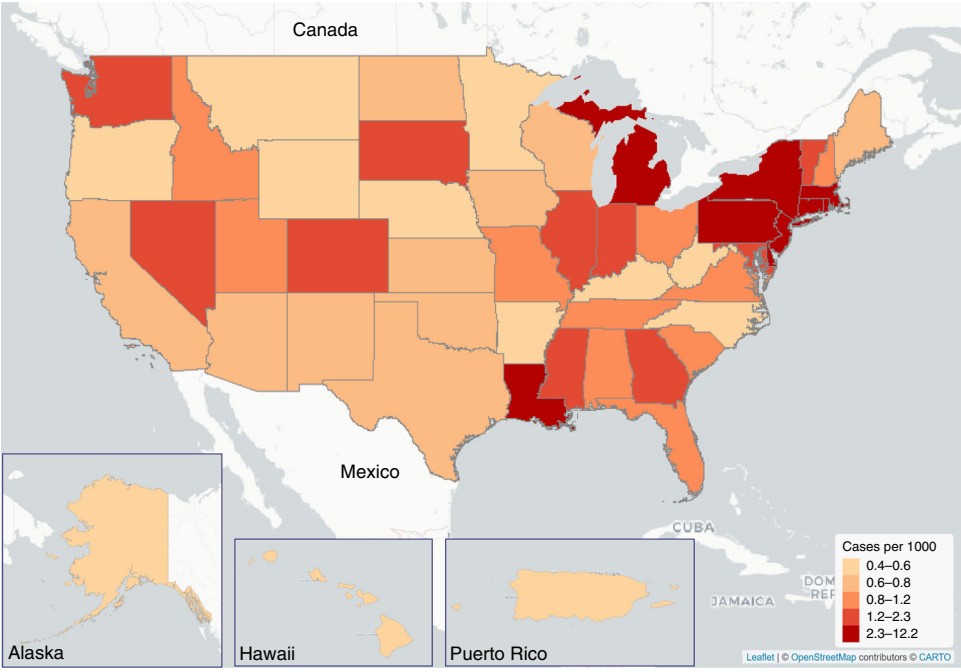

**b**  Ratio of cumulative estimated infections to confirmed COVID-19 cases

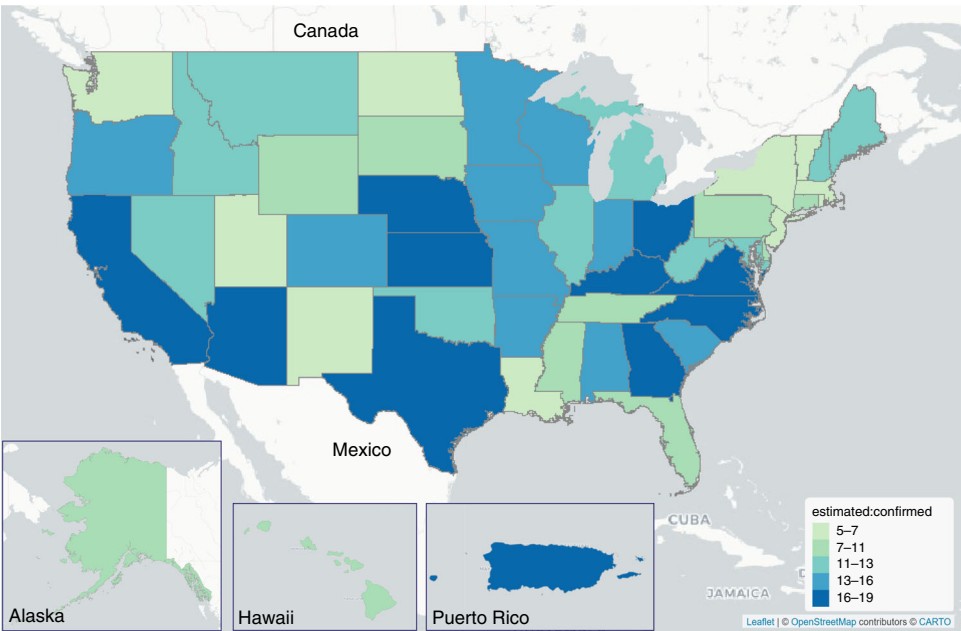

**Fig. 3 Map of confirmed COVID-19 case counts and the ratio of expected infections.** Confirmed COVID-19 cases per 1000 and estimated SARS-CoV-2 infections per 1000 varied by US state and region. Map of confirmed COVID-19 case counts and the ratio of expected infections. Each panel displays colors defined by quintiles of the distribution of **a** confirmed COVID-19 cases per 1000 and **b** the ratio of the median of the distribution of estimated infections from the probabilistic bias analysis to confirmed COVID-19 cases. Analyses include cumulative confirmed COVID-19 case counts up to April 18, 2020. Estimated infections were from a using semi-Bayesian probabilistic bias analysis to correct for incomplete testing and imperfect test accuracy. Estimated SARS-CoV-2 infections include both symptomatic and asymptomatic infections. Quality of daily estimates of the number of tests performed varied by state; see Supplementary Table 3 for details. Underlying map from OpenStreetMap available under the Open Database License (https://www.openstreetmap.org/copyright). See interactive plot at https://covid19epi.github.io/stats/.

Tracking Project: 721,245; CDC: 720,630). State-level reporting of tests completed and the number of positive tests by date varied; COVID Tracking Project assigned each state a data quality grade based on whether (1) reporting positives reliably, (2) reporting negatives sometimes, (3) reporting negatives reliably, and (4) reporting all commercial tests. Thirty-five states and the District of Columbia met all of these criteria, 14 states met three criteria, and one state met only two criteria

(Supplementary Table 3). We assumed that all test results included in this data source were done using polymerase chain reaction because during the study period, alternative tests (e.g., antibody tests) were not approved for diagnosis of SARS-CoV-2 infection by the U.S. Food and Drug Administration (such tests were only used for research purposes and are not likely to have been included in case counts). In addition, we assumed that the vast majority of samples collected were

nasopharyngeal swabs, which were recommended by the CDC as the preferred choice of swab[37].

**Overview of statistical methods.** Our aim was to estimate the total number of SARS-CoV-2 infections in each U.S. state from February 28 to April 18, 2020 using probabilistic bias analysis[22] to correct empirical confirmed case counts for bias due to incomplete testing and imperfect test accuracy. Using available evidence from the study period, we defined prior distributions of testing probabilities among individuals with moderate to severe symptoms requiring medical attention or hospitalization (e.g., shortness of breath, high fever) vs. those with minimal symptoms (e.g., cough without difficulty breathing or shortness of break, low grade fever) or no symptoms (see details below under "Prior definitions"). To quantify uncertainty in testing probabilities, we randomly sampled from each prior distribution $10^4$ times and estimated the total number of infections using a simple model relating the number of individuals tested to testing probabilities. Through this process we obtained a Monte Carlo estimate of the distribution of estimated SARS-CoV-2 infections in each state. We report the medians of the distribution of infections in each state and the simulation interval (2.5th and 97.5th percentiles). The estimated number of infections included confirmed COVID-19 cases and undiagnosed infections.

**Probabilistic bias analysis.** Bias analysis is a technique that attempts to correct for biases in observational data under assumptions about how the data are biased away from the true value (for example, due to selection bias or misclassification). When bias parameters that are used to correct for bias are treated as random variables with defined probability distributions, the procedure is known as probabilistic bias analysis. Probabilistic bias analysis is semi-Bayesian because it defines prior distributions for bias parameters but does not use a formal likelihood function to model the relation between these distributions and data[22]. In cases where a likelihood function is available yet data provide only limited information to update prior distributions, probabilistic bias analysis and fully Bayesian approaches lead to nearly identical results[38]. Given the significant uncertainty in several of the input parameters in this study, a fully probabilistic treatment may remain partially unidentified unless supplemented with very strong priors. Since our goal was to provide a more realistic picture of true infection burden that removed bias due to testing practices and varying diagnostic characteristics, we decided to use a transparent method that required fewer assumptions.

Because bias-corrected estimates often use a complex mathematical relationship to relate bias parameters and empirical data, analytic treatment of the induced probability distribution on the bias-corrected estimates is often intractable, and the distribution is investigated via Monte Carlo simulation. In order to correct observational case counts by state for selection bias (preferential testing of moderate-severe cases) and imperfect diagnostic accuracy, we developed a simple model based on epidemiologic formulae to incorporate testing and symptom probabilities following specified distributions into the final bias-corrected estimate (see details under "Correction for incomplete testing").

**Sampling prior distributions.** We defined prior distributions for seven parameters: P($S_1$ | tested), P($S_1$ | untested), P($S_0$ | test+), $\alpha$, $\beta$, SARS-CoV-2 test sensitivity and specificity based on available evidence (Table 1, Supplementary Fig. 3, Supplementary Table 4). We define $S_1$ to be an indicator variable of moderate to severe COVID-19 symptoms, and $S_0$ to be an indicator variable of minimal or no COVID-19 symptoms. P($S_1$ | tested) is the probability of having moderate to severe symptoms among tested individuals, and P($S_1$ | untested) is the analogous probability among untested individuals. P($S_0$ | test+) is the probability of having mild or no symptoms among individuals who tested positive. We defined $\alpha$ and $\beta$ as random variables describing the ratio of P(test + |$S_1$, untested) and P(test + |$S_0$, untested) to the empirical state-level estimate P(test + |tested).

For each state, we calculated the empirical estimate P(test+|tested) as the cumulative number of cases divided by the cumulative number of tests in each state from February 28, 2020 to April 18, 2020. Therefore P(test + |tested) is a point estimate of the average probability over this interval. We chose not to vary this quantity by date because the low testing rates per state resulted in unstable estimates of P(test+|tested), particularly when <1% of the population was tested, which was the case in most states until early-mid April (Supplementary Fig. 4). Furthermore, when restricting to dates when at least 0.6% of the population was tested in each state, P(test+|tested) was relatively stable within each state over time, suggesting that at least over the period considered, this quantity did not change significantly over time (Supplementary Fig. 5).

We sampled $10^4$ values from the distributions of P($S_1$ | tested), P($S_1$ | untested), $\alpha$, $\beta$, test sensitivity and test specificity. Using the state-level empirical estimates of P(test+ | tested) and sampled values of $\alpha$, $\beta$, we sampled P(test+ | $S_1$, untested) and P(test+ | $S_0$, untested), the probability an untested individual exhibiting moderate (likewise for none to mild) symptoms would test positive on their initial test. For untested individuals who had moderate to severe symptoms, we assumed that the probability they tested positive was 80–99% the empirical probability of testing positive among tested individuals in a state, P(test + |tested). Accordingly, $\alpha$ was defined such that $\alpha_0 < $ P(test + |$S_1$, untested)/P(test + |tested) $< \alpha_1$, with truncated support between $\alpha_0 = 80\%$ and $\alpha_1 = 99\%$. For untested individuals who had mild or no symptoms, we assumed that the probability they tested positive was 0.2–32% of the empirical probability. Likewise, $\beta$ was defined such that $\beta_0 < $ P(test + |$S_0$, untested)/P(test + |tested) $< \beta_1$, with truncated support on $\beta_0 = 0.2\%$ and $\beta_1 = 32\%$.

We assumed that the parameters, P($S_1$ | tested), P($S_1$ | untested), P($S_0$ | test+), $\alpha$, $\beta$, test sensitivity and test specificity were independent and identically distributed across states, whereas P(test + | tested) and consequently P(test + |$S_0$, untested) and P(test + |$S_1$, untested) varied between states based on empirical data. In principle, it is possible that some or all of these parameters were correlated between states. Between-state correlation in testing protocols for asymptomatic vs. symptomatic individuals may have existed; however, to our knowledge there is no evidence to inform assumptions about correlation between states during the study period. State-level correlation in testing probabilities (e.g., the probability that a mild or asymptomatic individual was tested) could affect test sensitivity as lower viral loads are less likely to be detected[39]. However, there is evidence that asymptomatic and symptomatic individuals have similar viral loads, and as such, we would not expect sensitivity to vary by state during the study period[40,41]. During the early epidemic period, there was also no evidence that laboratory practices or the type of sample collected for RT-PCR (e.g., nasopharyngeal vs. throat swabs), both of which could affect diagnostic accuracy, varied substantially by state. To our knowledge, there is no available data for the study period that could be used to reasonably assess between-state correlation in parameters to estimate an appropriate correlation matrix. For these reasons, we assumed parameters were uncorrelated between states, with the caveat that if this assumption is incorrect, simulation intervals for estimated cumulative infection burden will be too narrow.

**Constraining of prior distributions.** The quantity P($S_0$ | test+) is a function of other parameters P($S_1$ | untested), $\alpha$, $\beta$, meaning its distribution is entirely determined by those three parameters. However, we considered estimates of P($S_0$ | test +) to be more reliable than those of P(test + |$S_1$) and P(test + | $S_0$) because P($S_0$ | test +) has been estimated in numerous published studies, including some with more representative sampling[1,2,4,42–45]. Thus, we used Bayesian melding to incorporate both sources of information into a joint (melded) distribution on P($S_0$ | test +) and P($S_1$ | test +) = 1 – P($S_0$ | test +)[46]. Because the distribution of P($S_0$ | test +) is constrained to take on plausible values in accordance with other parameters, it is unlikely that the priors disagree strongly with observed data, and probabilistic bias analysis should resemble a fully Bayesian approach[38].

**Table 1 Prior distributions for probabilistic bias analysis.**

| Distribution | Minimum (lower bound) | Mean | Maximum (upper bound) | Shape 1 | Shape 2 |
|---|---|---|---|---|---|
| P($S_1$|tested) | 0.00 | 0.93 | 1.00 | 20.00 | 1.40 |
| P($S_1$|untested) | 0.00 | 0.03 | 0.15 | 1.18 | 45.97 |
| $\alpha$ | 0.80 | 0.90 | 1.00 | 49.73 | 5.53 |
| $\beta$ | 0.002 | 0.15 | 0.40 | 2.21 | 12.53 |
| P($S_0$|test+) | 0.25 | 0.42 | 0.70 | 6.00 | 9.00 |
| Sensitivity ($S_e$) | 0.65 | 0.86 | 1.00 | 4.20 | 1.05 |
| Specificity ($S_p$) | 0.9998 | 0.9999 | 1.00 | 4998.50 | 0.25 |

P($S_1$|tested) is the probability of having moderate to severe symptoms among tested individuals, and P($S_1$|untested) is the analogous probability among untested individuals. We defined $\alpha$ and $\beta$ as random variables describing the ratio of P(test + |$S_1$, untested) and P(test + |$S_0$, untested) to the empirical state-level estimate P(test + |tested). P($S_0$|test+) is the probability of having mild or no symptoms among individuals who tested positive. Detailed descriptions of each prior distribution including cited literature are given in the Methods section. For truncated Beta distributions (those with lower and upper bounds not equal to 0 and 1), the mean was calculated via numerical integration. Distributions truncated to region ($a$,$b$) are defined by modifying the untruncated density function $f(x)$ to be: $\frac{f(x)}{F(b)-F(a)}$, where $F(x)$ is the distribution function, such that the truncated density integrates to 1 over that region. The values for P($S_0$|test+) give the distribution prior to Bayesian melding.

The relationship between $\varphi = P(S_0 \mid \text{test} +)$ and the parameters $\theta = \{P(S_1 \mid \text{untested}), \alpha, \beta\}$ is given by a function $(M: \theta \rightarrow \varphi)$:

$$P(S_0 \mid \text{test} +) = \frac{\beta(1 - P(S_1 \mid \text{untested}))}{\beta(1 - P(S_1 \mid \text{untested})) + \alpha P(S_1 \mid \text{untested})} \quad (1)$$

Mathematically, once a distribution is assigned to $\theta$ and $M$ is defined, the distribution of $\varphi$ is fully specified. Bayesian melding[46], allows one to combine prior distributions on function inputs $\theta$ and output $\varphi$ to generate a joint melded prior distribution on $\{\theta, \varphi\}$. We assumed that $P(S_0 \mid \text{test} +)$ followed a truncated beta distribution (defined below) which was combined with $\theta = \{\alpha, \beta, P(S_1 \mid \text{untested})\}$ and the function $M$ given in Eq. (1) to produce the final distribution of $P(S_0 \mid \text{test} +)$, shown in Supplementary Fig. 3. The advantage of this method is that it incorporates all available prior uncertainty as well as constraints on plausible output given by $M$. To sample values of $\{\theta, \varphi\}$ to be used as input to the probabilistic bias correction, we sampled $10^5$ variates from the melded prior distribution using a Sampling-Importance-Resampling algorithm[46]. We also sampled $P(S_1 \mid \text{test} +) = 1 - P(S_0 \mid \text{test} +)$ using the same procedure.

**Correction for incomplete testing.** Sampled variates from the procedure described above were used as inputs to the bias correction. To correct for incomplete testing, we defined the following formulas to estimate, for each state, the number of SARS-CoV-2 infections among untested individuals. $N_{\text{untested},S_1}^+$ is the estimated number of untested individuals who would have moderate to severe symptoms and test positive if tested. $N_{\text{untested},S_0}^+$ is the estimated number of untested individuals who would have mild or no symptoms and test positive if tested.

$$N_{\text{untested},S_1}^+ = P(S_1 \mid \text{untested}) \times P(\text{test} + \mid S_1, \text{untested}) \times N_{\text{untested}} \quad (2)$$

$$N_{\text{untested},S_0}^+ = (1 - P(S_1 \mid \text{untested})) \times P(\text{test} + \mid S_0, \text{untested}) \times N_{\text{untested}} \quad (3)$$

The quantities $N_{\text{untested},S_1}^+$ and $N_{\text{untested},S_0}^+$ are, marginally, binomial random variables, with parameters $n = N_{\text{untested}}$ and probability parameter $P(\text{test}+, S_1, \text{untested}) = P(\text{test} + \mid S_1, \text{untested}) \times P(S_1 \mid \text{untested}) \times P(\text{untested})$. In our analysis, we set them equal to their expectation because the binomial variance is negligible in comparison to other sources of uncertainty, and the population size is large enough for the random variable to be highly concentrated around its expectation value. Incorporation of this variation would simply involve additional resampling at this stage.

**Correction for imperfect test accuracy.** Finally, to estimate the number of SARS-CoV-2 infections ($N^*$) correcting for imperfect test accuracy (i.e., sensitivity ($S_e$) or specificity ($S_p$) < 1), we used the following formula[47]:

$$N^* = (N^+ - ((1 - S_p) \times N)) / (S_e + S_p - 1) \quad (4)$$

where $N^+$ is the sum of the number of confirmed cases ($N_{\text{tested}} = N_{\text{tested},S_1}^+ + N_{\text{tested},S_0}^+$) and the estimated number of infections among untested individuals correcting for incomplete testing ($N_{\text{untested}}^+ = N_{\text{untested},S_1}^+ + N_{\text{untested},S_0}^+$).

Supplementary Fig. 6 presents a visual depiction of the sampling procedure described above.

**Simulation results.** To fully propagate uncertainty into the final state-specific estimates of case counts $N^*$, we repeated the process described above $10^4$ times, obtaining a distribution of expected cases in each state on each day as the primary estimate. To characterize uncertainty in our model, we report the 2.5th and 97.5th quantiles of the sampled distributions.

**Proportion of infections attributable to under-testing.** We calculated the proportion of the difference between observed case counts and the estimated number of infections attributable to imperfect test accuracy as the difference between $N^*$ with $S_e$ and $S_p$ set to our prior values and with $S_e = 1$ and $S_p = 1$ divided by the difference between $N^*$ and the observed case counts. We calculated the proportion attributable to incomplete testing as the $1 - $ the proportion attributable to imperfect test accuracy. To obtain national estimates of these percentages, we obtained the median and 2.5th and 97.5th quantiles of the state-specific distributions and weighted by state population.

$$\text{Proportion attributable to imperfect testing} = \frac{N^* - N^+}{N^* - N_{\text{tested}}^+} \quad (5)$$

$$\text{Proportion attributable to incomplete testing} = 1 - \left(\frac{N^* - N^+}{N^* - N_{\text{tested}}^+}\right) \quad (6)$$

**Sensitivity analyses.** To assess the robustness of our assumptions to plausible alternative priors, we defined seven alternative parameterizations of the joint prior distribution and repeated analyses for each scenario (Supplementary Table 2). Scenarios 1–4 shifted distributions of parameters that had the least available

published evidence to support our prior specifications ($\alpha$, $\beta$, $P(S_1 \mid \text{tested})$, $P(S_1 \mid \text{untested})$). Scenarios 6–7 assessed the robustness of our assumption that $\alpha$ and $\beta$ are independent. We sampled $10^4$ values of estimated cumulative infection for each state under each scenario.

1. Moderately lower test positivity among untested individuals with moderate to severe symptoms: We reduced the lower bound of truncation for the distribution of $\alpha$ from 0.80 to 0.50 and reduced the mean from 0.90 to 0.85.
2. Substantially lower test positivity among untested individuals with moderate to severe symptoms: We reduced the lower bound of truncation for the distribution of $\alpha$ from 0.80 to 0.25 and reduced the mean from 0.90 to 0.75.
3. Lower probability of being symptomatic among those tested: We shifted the mean of the distribution of $P(S_1 \mid \text{tested})$ from 0.93 to 0.80.
4. Higher probability of being symptomatic among those not tested: We increased the upper bound of the distribution of $P(S_1 \mid \text{untested})$ from 0.15 to 0.25.
5. Higher test positivity among untested individuals with mild to no symptoms: We increased the upper bound of the distribution of $\beta$ from 0.40 to 0.60 and the mean was shifted up from 0.15 to 0.25.
6. Mild correlation between $\alpha$, $\beta$. In order to test robustness of results to assumptions of independence between $\alpha$ and $\beta$ we used a Gaussian copula to simulate correlation between $\alpha$, $\beta$ with $\rho = 0.2$. The marginal distributions were not changed.
7. High correlation between $\alpha$, $\beta$. We simulated $\alpha$, $\beta$ from a Gaussian copula with $\rho = 0.8$. The marginal distributions were not changed.

**Definition of prior distributions.** To define prior distributions, we reviewed the available literature on testing probabilities and diagnostic accuracy and used the best available evidence specific to the study period. Our approach to defining prior distributions is evidence-based and transparent given the limitations of relevant published studies at this time. Given the limited available data on testing probabilities during the study period, in defining prior distributions we erred in favor of greater uncertainty; in most cases (except specificity) where we felt a concentrated prior was appropriate we intentionally made the distribution more diffuse, giving more weight to parameter values farther from the mean and therefore encompassing more potential for extreme scenarios. Thus, the width of our prior distributions reflects our genuine uncertainty about each prior distribution.

**Definition of distribution of $P(S_1 \mid \text{tested})$.** We defined the distribution of $P(S_1 \mid \text{tested})$ as a truncated beta distribution with the bulk of the distribution between 60 and 100% (mean: 93.4%) under the assumption that the vast majority of individuals tested in the U.S. in March 2020 had moderate to severe symptoms. March 4, 2020, the CDC instructed physicians to use their judgment to determine which patients to test for SARS-CoV-2. Given the limited availability of tests in the U.S., they advised that symptomatic patients were prioritized for testing. On March 24, 2020, the CDC recommended that testing be prioritized for hospitalized patients, symptomatic health care workers, patients in vulnerable populations, and individuals with mild symptoms in communities experiencing a large number of COVID-19 hospitalizations[37]. State-level testing priorities mostly followed CDC recommendations (Supplementary Table 1).

Our prior range is consistent with the findings of the small number of studies that have found that the majority of individuals tested for SARS-CoV-2 symptoms in hospitals in the U.S. had COVID-19 symptoms. A review of emergency department admissions for COVID-19 in San Diego, California from March 10–19, 2020 found that 10% of patients had fever upon arrival, and 65% of patients were classified as urgent, emergency, or resuscitation, indicating their severity at the time of testing and admission. 90.6% had symptoms[48]. An analysis of individuals tested at the University of Utah Health hospital from March 10–April 24, 2020, a period when Utah had expanded testing capacity, reported that 89% of tested individuals had a cough, 65% had a fever, 64% had shortness of breath[49]. A study of COVID-19 hospital admissions for COVID-19 at a hospital in the Des Moines from March 1 to April 4, 2020 reported that 94% had history of fever, 88% dry cough, 81% dyspnea, 25% athralgia, 25% headache, 19% anosmia, and 19% loss of taste[50].

**Definition of distribution of $P(S_1 \mid \text{untested})$.** We defined the distribution of $P(S_1 \mid \text{untested})$ as a truncated beta distribution with a range from 0 to 15% (mean: 2.5%). A SARS-CoV-2 study in Iceland that enrolled a random sample of individuals reported that 0.9% all individuals reported fever, 3.8% reported cough, and 0.6% reported a loss of smell or taste[42]. The symptom profile of these individuals likely represents that of the greater population given the low proportion who tested positive for SARS-CoV-2. While not directly comparable, studies of influenza symptoms, which overlap with COVID-19 symptoms, provide an estimate of the general frequency of fever and cough in the population. Since the majority of the population tested for SARS-CoV-2 in the U.S. had moderate to severe symptoms, these probabilities provide a reasonable estimate of $P(S_1 \mid \text{untested})$. Cohort studies of influenza-like illness (fever and cough and/or sore throat) in prior years in the U.S. and the Netherlands have found weekly incidence of 1–4%[51–53]. Studies measuring over the entire influenza season have reported cumulative incidence of influenza-like illness ranging from ~7 to 17%[54–56].

**Definition of distribution of $\alpha$.** To allow for state-level variation in P(test $+|S_1$, untested) due to differing transmission dynamics in each state, our model allowed this probability to vary by state. We defined $\alpha$ as a truncated beta distribution ranging from 80 to 100% (mean: 90%) such that $\alpha_0 <$ P(test $+|S_1$, untested)/P(test $+|$tested) $< \alpha_1$. A study of COVID-19 patients at a hospital in Wuhan, China in January–February 2020 tested individuals with fever, cough, or hard breath or individuals in close contact with COVID-19 patients. They reported higher test positivity among patients who presented with fever compared to all patients tested. While this study does not provide direct estimates of this prior, it demonstrates the range of test positivity in one city among individuals with a range of symptom presentation[57]. We used an empirical state-specific estimate of P(test $+|$ tested) equal to the cumulative number of cases divided by the cumulative number of tests by April 18, 2020. The median of state-specific distributions of P(test $+|S_1$, untested) ranged from 0.02 to 0.46, and in many states the medians were close to 0.1 (Supplementary Table 4). This is consistent with a study in Iceland, which estimated that P(test $+|S_1$) was 1% among those tested through population screening ($N = 3579$) and 2% among those tested in a random sample ($N = 271$)[42].

**Definition of distribution of $\beta$.** To allow for state-level variation in P(test$+|S_0$, untested) due to differing transmission dynamics in each state, our model allowed this probability to vary by state. We defined $\beta$ as a truncated beta distribution ranging from 0.2 to 40% (mean: 15%) such that $\beta_0 <$ P(test $+|S_0$, untested)/P(test $+|$tested) $< \beta_1$. As described above, we used an empirical state-specific estimate of P(test $+|$ tested). The median of state-specific distributions of P(test $+|S_0$, untested) ranged from 0.001 to 0.023 (Supplementary Table 4).

Only one study has collected data from a population sample to inform this prior: a study in Iceland estimated that P(test $+|S_0$) was 0.8% among mild symptomatic/asymptomatic individuals tested in population screening and 0.6% ($N = 10,797$) among mild symptomatic/asymptomatic individuals tested in a random sample ($N = 2283$)[42]. Studies testing health workers and pregnant women at the time of admission for delivery in the early phase of the pandemic for SARS-CoV-2 shed light on this prior even though these populations are not generally representative. Three studies estimated test positivity among health care workers who were mildly symptomatic or asymptomatic. Test positivity in these studies was 11% among mildly symptomatic workers in mid-late March 2020 in the Netherlands ($N = 803$)[58], 18% among mildly symptomatic workers in mid-March 2020 in the United Kingdom ($N = 1533$)[59], and 0.9% among asymptomatic medical staff in January and early February 2020 Wuhan, China ($N = 335$)[60]. Two studies of pregnant women conducted in late March and early April 2020 estimated test positivity among pregnant women admitted for delivery, regardless of symptoms. A study in New York City screening 215 pregnant women admitted for delivery, regardless of symptoms, found that 13.7% of women without symptoms tested positive[61]. A study of 756 pregnant women in Connecticut found that 2.9% of women who were asymptomatic tested positive[62]. However, these estimates may not be representative of the general population because of differences in age and immunity of pregnant women. The upper bound of P(test $+|S_0$, untested) in states with high test positive probabilities includes these percentages (Supplementary Table 4).

**Definition of distribution of P($S_0|$test$+$).** We defined the distribution of P($S_0|$test $+$) as a truncated beta distribution with a range from 25 to 70% (mean: 42%) because studies that have tested both symptomatic and asymptomatic individuals have found estimated probabilities within this range[1,2,4,42,43]. In these studies, it is possible that individuals who were asymptomatic at the time of testing developed symptoms later. A meta-analysis reported that 25.9% (95% CI: 18.8%, 33.1%) of individuals who tested positive were asymptomatic at the time of diagnosis ($N = 25$ studies)[45], and a narrative review reported a similar range of estimates from 16 studies[44]. Though there were more studies to support this prior than for other priors, we chose not perform a meta-analysis to obtain the prior distribution due to limitations that affect the generalizability of the majority of these studies. These include very small sizes (nearly 31% in the meta-analysis[45] had a sample size ≤10) and enrollment of non-representative populations, such as pregnant women, employees at a specific location, travelers on cruise ships, and nursing home residents. A recent narrative review which included many of the same studies came to similar conclusions regarding the risks of pooling data for formal analysis[44].

**Definition of distribution of test sensitivity ($S_e$).** We defined the distribution of SARS-CoV-2 test sensitivity as a truncated beta distribution with a range from 65 to 100% (median: 87%) because the available evidence to date has reported clinical sensitivity within this range. The U.S. CDC RT-PCR test has analytical sensitivity ≥95% for RNA concentrations ≥1 copy $\mu L^{-1}$ [63]. A study of 213 RT-PCR-confirmed COVID-19 patients from Wuhan reported that the detection rate in the first 14 days since onset was 54–73% for nasal swabs and 30–61% for throat swabs[9]. Using any RT-PCR as gold standard, another study of 1014 patients from three Chinese provinces found that initial RT-PCR pharyngeal sensitivity ranged from 65 to 80%[11]. Another small study ($N = 4$) of COVID-19 patients from Wuhan also found that some that had been discharged tested positive 5–13 days later, suggesting that initial tests produced false negatives[12]. Test positivity is highest soon after symptom onset and declines subsequently and is close to zero more than

35 days after symptom onset; it is likely that test sensitivity follows a similar pattern[64]. Our prior for test sensitivity assumes testing was conducted within the first 2 weeks of disease.

**Definition of distribution of test specificity ($S_p$).** We defined the distribution of SARS-CoV-2 test specificity as a truncated beta distribution with a range from 99.98 to 100% (median: 99.99%). The SARS-CoV-2 RT-PCR diagnostic panel developed by the CDC was designed to minimize the chance of a false positive, so test specificity assumed to be very high in laboratories that comply with standard best practices, such as the use of negative controls[63]. Validation of both the CDC and WHO real-time polymerase chain reaction (PCR) tests for SARS-CoV-2 found no false positives in cell culture samples containing other respiratory viruses[37,65].

**Reporting summary.** Further information on research design is available in the Nature Research Reporting Summary linked to this article.

## Data availability
The data that support the findings of this study are available from the COVID Tracking Project (https://covidtracking.com/).

## Code availability
All code necessary to reproduce the entirety of findings of this study have been made publicly available at the GitHub repository (https://github.com/jadebc/covid19-infections/releases/tag/NatureComms) and a permanent archive of the version of the code has been made at https://doi.org/10.5281/zenodo.3976252. The file "README.md" in the root directory of the repository contains instructions on how to recreate all presented figures and tables in the manuscript.

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

## Acknowledgements

The authors thank Amy J. Markowitz, JD for editorial assistance with this paper. This study was supported by Flu Lab. The funders of the study had no role in the study design, data analysis, data interpretation, or writing of the paper.

## Author contributions

Conceptualization: J.B.-C. Data curation: A.N., N.N.P., and S.D. Formal analysis: J.B.-C., A.N., and S.L.W. Funding acquisition: J.B.-C. Investigation: J.B.-C., Y.S.C., and A.N.M. Methodology: J.B.-C., A.N., S.L.W., and A.H. Project administration: J.B.-C. Software: A.N., N.N.P., S.D., A.S., A.N.M., and S.L.W. Supervision: J.B.-C. Validation: A.N. and A.N.M. Visualization: J.B.-C., A.N., N.N.P., S.D., A.S., and S.L.W. Writing—original draft preparation: J.B.-C. and S.L.W. Writing—review and editing: J.B.-C., S.L.W., Y.S.C., A.N.M., M.S.H., J.M.C., A.R., B.F.A., and A.H.

## Competing interests

The authors declare no competing interests.
