## [Peer Review File · Nature Communications]

REVIEWER COMMENTS

Reviewer #1 (Remarks to the Author):

I thank the authors for responding so thoroughly to my previous review. The paper is a big improvement.

Table 1 still appears to be incorrect: the prior for the specificity has a mean that lies to the left of the lower bound. Could the authors provide the mathematical form for these priors? I had imagined from their first submission that $p' = a + (b-a)p$ with $p \sim \text{be}(c,d)$ but it appears not. So how actually are these implemented. I am much confused.

The structure of the manuscript is forced, because the requirement to put the methods at the end combined with the authors' natural desire to explain what they're doing means that more than half of the results section is, actually, methods, with another set of methods at the end of the paper. If the journal's policy on structure is really immutable, then my recommendation to the authors would be to move Methods Overview and Prior Definitions out of the Results section and into the Methods at the end, reintegrate it with the text there, and let the reader read the intro, skip to the end for the methods, and come back to the results like a normal journal article.

The mini-methods section in the results section says the approach is conservative. This is one of the most misused terms in scientific writing, it suggests a bias but a bias in a way that is okay because it's in a safe direction, but which direction? Are your estimates safely low, or safely high? What exactly makes them conservative? Please justify.

The authors write that

"In each state, we estimated the probability that untested individuals would have tested positive had they been tested. These priors capture the test positivity we expect if, counter to fact, SARS-CoV-2 tests had been universally available and performed regardless of symptom presentation." How is time accounted for in this? Are you imagining daily testing, or a one-off sweep? The examples you provide (from random and screening samples, from HCWs and from pregnant women) are misleading without giving context about the epidemic state at the time point the studies were done. Please clarify.

When you write that the proportion of minimal symptoms is well estimated from other studies, why not meta analyse those studies to derive an objective prior distribution, rather than use a subjectively selected one? I would have thought that using objective ingredients when available would be preferable to 'making distributions up'.

When you define your seven parameters in the methods, can this be done a little more carefully please? Clarify that $P(\text{test} + | S1, \text{***untested***})$ means on first usage. The relationships would probably be a bit clearer if they were spelled out as formal equations.

Don't start a sentence with a lower case letter. So: " $\phi = \dots$ is related to the parameters θ ... given by" would be better as "The relationship between $\phi = \dots$ and the parameters θ ... is given by" so as not to raise the hackles of pedants.

Missing comma before 'marginally' in the sentence beginning "The quantities $N_{\{\text{untested}, S_1\}^{\{+\}}$ and ... are marginally, binomial"

"we set them equal to their expectation values"  "to their expected values" and similarly later in the paragraph. Or just write 'expectation' since this is already a value and a noun modifying another noun can make non-Americans cringe.

Would it be better for the sensitivity and specificity to have subscripts rather than being two letters together? Eg S_{e_1} for the former rather than S_e , which might be confused with a product of S and

e.

For figure 2, I wonder whether you would want to drop the confirmed cases from 2A (because it is truly hard to make both bars out). You already provide the ratio in 2B so the information is there, but the message of the plot would be more impactful, I think, if the data could be more clearly seen. Also, two further suggestions: consider increasing the vertical size of the plot so that the state names don't abut each other quite so much, and would it be worth maintaining the same order on the two panels (probably by panel A) so that differences can be more readily apparent?

Figure 3: No name for Canada eh?

Appendix F2: Thank you for responding to my point. I don't think box plots with outliers make sense for distributions of parameters. The calculations for outliers in datasets, sure, that makes sense. But for a parameter distribution, there ought not to be outliers unless you didn't sample enough. May I suggest you either (1) drop the 'outlier' points and extend the whiskers to the 2.5th and 97.5th percentiles? Or (2) consider a distributional plot like a violin plot. Also if this is for the appendix, could you make the fonts for the axis labels bigger and make the plot longer to accommodate? Not everyone is young and able to read tiny fonts. (Same for the appendix figure 5)

Reviewer #2 (Remarks to the Author):

The authors have made a number of improvements to the earlier version of the paper in response to my comments and those of other reviewers. The description of the methods is generally improved and the paper is strengthened the addition of at least some sensitivity analyses.

One or two concerns remain. The results of the paper remain quite unsurprising perhaps – mainly that it is not possible with any certainty to ascertain the total number of covid-19 cases and, consequently the degree of herd immunity.

Some more specific comments are given below:

P. 8 The parameters alpha and beta are introduced before they are defined and the current definition remains unclear. It would be useful to define these in the text before first use. One has to infer the definition of these are the ratios appearing between the upper and lower bounds α_0 , α_1 , and β_0 and β_1 .

Why should the test sensitivity and specificity be uncorrelated across states (p, 8)? This seems like a very implausible assumption. The prior you place on these parameters represents your belief about the accuracy of the test. Undoubtedly positive correlations between sensitivity across states would lead to a even greater uncertainty in the total number of cases at the national level? More discussion of this point, and justification of the independence assumption should be given.

Table 2 – are you not conditioning on being untested also here i.e. $P(\text{test} + |S_0, \text{untested})$? If so, there is an inconsistency between what is stated in the text and the table which should be corrected.

We found the peer reviewer comments to our initial submission to be positive and very constructive. My co-authors and I are pleased to submit a revised manuscript that addresses the Referees' suggestions as well as point-by-point responses to Referee comments. We believe that this revised version is greatly improved.

Reviewer #1 (Remarks to the Author):

I thank the authors for responding so thoroughly to my previous review. The paper is a big improvement.

- 1. Table 1 still appears to be incorrect: the prior for the specificity has a mean that lies to the left of the lower bound. Could the authors provide the mathematical form for these priors? I had imagined from their first submission that $p' = a + (b-a)p$ with $p \sim \text{be}(c,d)$ but it appears not. So how actually are these implemented. I am much confused.**

Response: We have fixed our omission of several decimal points in Table 1 that made it appear the mean was below the truncation bound. We also added the following text to the table caption to make it clear how the truncated distribution was defined:

"Distributions truncated to region $(a,b]$ are defined by modifying the untruncated density function $f(x)$ to be: $\frac{f(x)}{F(b)-F(a)}$, where $F(x)$ is the distribution function, such that the truncated density integrates to 1 over that region."

- 2. The structure of the manuscript is forced, because the requirement to put the methods at the end combined with the authors' natural desire to explain what they're doing means that more than half of the results section is, actually, methods, with another set of methods at the end of the paper. If the journal's policy on structure is really immutable, then my recommendation to the authors would be to move Methods Overview and Prior Definitions out of the Results section and into the Methods at the end, reintegrate it with the text there, and let the reader read the intro, skip to the end for the methods, and come back to the results like a normal journal article.**

Response: We agree, and we discussed this feedback with the Editor, who recommended that we move the Methods Overview and Prior Definitions to the Methods section. The Results section now focuses on results only and is much more succinct. We integrated the Methods Overview and Prior Definitions into the Methods section as suggested.

- 3. The mini-methods section in the results section says the approach is conservative. This is one of the most misused terms in scientific writing, it suggests a bias but a bias in a way that is okay because it's in a safe direction, but which direction? Are your estimates safely low, or safely high? What exactly makes them conservative? Please justify.**

Response: In response to comment #2, we relocated this sentence to the Methods section under “Prior Definitions”. We have removed the word “conservative” from this sentence, and we clarified the language in this section to explain that, for all priors except specificity, we intentionally made the distribution more diffuse. This means that more probability mass is given to values of those parameters farther away from the mean or modal value (because the Beta distributions we use are all unimodal, those two values will be nearly identical). We revised the text in the Methods section under “Definition of prior distributions” (page 11) accordingly:

“Given the limited available data on testing probabilities during the study period, in defining prior distributions we erred in favor of greater uncertainty; in most cases (except specificity) where we felt a concentrated prior was appropriate we intentionally made the distribution more diffuse, giving more weight to parameter values farther from the mean and therefore encompassing more potential for extreme scenarios. Thus, the width of our prior distributions reflects our genuine uncertainty about each prior distribution.”

4. The authors write that “In each state, we estimated the probability that untested individuals would have tested positive had they been tested. These priors capture the test positivity we expect if, counter to fact, SARS-CoV-2 tests had been universally available and performed regardless of symptom presentation.” How is time accounted for in this? Are you imagining daily testing, or a one-off sweep? The examples you provide (from random and screening samples, from HCWs and from pregnant women) are misleading without giving context about the epidemic state at the time point the studies were done. Please clarify.

Response: In describing how $P(\text{test+} \mid \text{tested})$ and, consequently, how $P(\text{test+} \mid S_0, \text{untested})$ and $P(\text{test+} \mid S_1, \text{untested})$ are calculated, we clarified the text to make clear that our estimate of $P(\text{test+} \mid \text{tested})$ is a simple average of daily data from the early epidemic period (February 28 to April 18, 2020). We chose to do this because in the early stage of the pandemic, the number of tests performed each day was relatively low. As a result, the test positivity rate among tested individuals was very unstable, as shown in Appendix Figure 3. The new text in the Methods section under “Sampling prior distributions” (page 7) reads:

“For each state, we calculated the empirical estimate $P(\text{test+} \mid \text{tested})$ as the cumulative number of cases divided by the cumulative number of tests in each state from February 28, 2020 to April 18, 2020. Therefore $P(\text{test+} \mid \text{tested})$ is a point estimate of the average probability over this interval.”

Regarding the timing of studies that measured health care workers and pregnant women, we have added information about the timing of these studies, which all took place in the early phase of the pandemic. The following text was modified in the Methods section on page 13:

“Studies testing health workers and pregnant women at the time of admission for delivery in the early phase of the pandemic for SARS-CoV-2 shed light on this prior, even though these populations are not generally representative. Test positivity in these studies was 11% among

mildly symptomatic workers in mid-late March 2020 in the Netherlands (N=803),⁵⁷ 18% among mildly symptomatic workers in mid-March 2020 in the United Kingdom (N=1,533),⁵⁸ and 0.9% among asymptomatic medical staff in January and early February 2020 Wuhan, China (N=335).⁵⁹ Two studies of pregnant women conducted late March and early April 2020 estimated test positivity among pregnant women admitted for delivery, regardless of symptoms. A study in New York City screening 215 pregnant women admitted for delivery, regardless of symptoms, found that 13.7% of women without symptoms tested positive.⁶⁰ A study of 756 pregnant women in Connecticut found that 2.9% of women who were asymptomatic tested positive.⁶¹”

5. When you write that the proportion of minimal symptoms is well estimated from other studies, why not meta analyse those studies to derive an objective prior distribution, rather than use a subjectively selected one? I would have thought that using objective ingredients when available would be preferable to ‘making distributions up’.

Response: While multiple studies have estimated $P(S_0|test +)$ and there is a preprint of a meta-analysis of these studies during the early epidemic period (Koh et al., 2020), the majority of published studies and preprints have substantial limitations that limit their generalizability. These include extremely small (< 10) sample sizes and enrollment of non-representative populations, such as pregnant women, employees at a specific location, cruise ships, and nursing homes. We feel that the quality of estimates available for our study period are not suitable for formal meta-analysis. We note that a recent narrative review on this topic arrived at the same conclusion and chose not to meta-analyze studies on this topic (Oran et al., 2020). We added the following text to the paragraph “Distribution of $P(S_0|test +)$ ” (page 13) to explain our rationale for this choice:

“Though there were more studies to support this prior than for other priors, we chose not perform a meta-analysis to obtain the prior distribution due to limitations that affect the generalizability of the majority of these studies. These include very small sizes (nearly 31% in the meta-analysis⁴² had a sample size ≤ 10) and enrollment of non-representative populations, such as pregnant women, employees at a specific location, travelers on cruise ships, and nursing home residents. A recent narrative review which included many of the same studies came to similar conclusions regarding the risks of pooling data for formal analysis.⁴⁵”

42 Koh WC, Naing L, Rosledzana MA, et al. What do we know about SARS-CoV-2 transmission? A systematic review and meta-analysis of the secondary attack rate, serial interval, and asymptomatic infection. *medRxiv* 2020; 2020.05.21.20108746.

45 Oran DP, Topol EJ. Prevalence of Asymptomatic SARS-CoV-2 Infection. *Annals of Internal Medicine* 2020; published online June 3. DOI:10.7326/M20-3012.

6. When you define your seven parameters in the methods, can this be done a little more carefully please? Clarify that $P(test + | S_1, \text{*untested***})$ means on first usage.**

The relationships would probably be a bit clearer if they were spelled out as formal equations.

Response: We have clarified that probability refers to the probability of that result on the untested person's initial test. We have restructured the paragraph where its definition appears and added text to clarify how that value is related to prior distributions and empirical data from states. Additionally, we have moved the paragraph which describes $P(\text{test+} \mid \text{tested})$ to precede this paragraph as those estimates are required to compute $P(\text{test+} \mid S_1, \text{untested})$ and $P(\text{test+} \mid S_0, \text{untested})$. These changes are in the Methods section under "Sampling prior distributions" on page 7. The specific sentence that address the first usage issue is as follows:

"Using the state-level empirical estimates of $P(\text{test+} \mid \text{tested})$ and sampled values of α, β , we sample $P(\text{test+} \mid S_1, \text{untested})$ and $P(\text{test+} \mid S_0, \text{untested})$, the probability an untested individual exhibiting moderate to severe symptoms (likewise for none to mild) would test positive on their initial test."

7. Don't start a sentence with a lower case letter. So: " $\phi=...$ is related to the parameters $\theta...$ given by" would be better as "The relationship between $\phi=...$ and the parameters $\theta...$ is given by" so as not to raise the hackles of pedants.

Response: We have made this change to the sentence where it appeared.

8. Missing comma before 'marginally' in the sentence beginning "The quantities $N_{\{\text{untested}, S_1\}^{\{+\}}$ and ... are marginally, binomial"

Response: We have made these changes.

9. "we set them equal to their expectation values"  "to their expected values" and similarly later in the paragraph. Or just write 'expectation' since this is already a value and a noun modifying another noun can make non-Americans cringe.

Response: We have made these changes.

10. Would it be better for the sensitivity and specificity to have subscripts rather than being two letters together? Eg S_e for the former rather than Se , which might be confused with a product of S and e .

Response: Thank you for the suggestion, we have made this change throughout the text as well as in figures where necessary.

11. For figure 2, I wonder whether you would want to drop the confirmed cases from 2A (because it is truly hard to make both bars out). You already provide the ratio in 2B so the information is there, but the message of the plot would be more impactful, I think, if the data could be more clearly seen. Also, two further suggestions: consider

increasing the vertical size of the plot so that the state names don't abut each other quite so much, and would it be worth maintaining the same order on the two panels (probably by panel A) so that differences can be more readily apparent?

Response: Thank you for the suggestion, we have made this change to Figure 2.

12. Figure 3: No name for Canada eh?

Response: Thank you for the suggestion, we have made this change to Figure 3.

13. Appendix F2: Thank you for responding to my point. I don't think box plots with outliers make sense for distributions of parameters. The calculations for outliers in datasets, sure, that makes sense. But for a parameter distribution, there ought not to be outliers unless you didn't sample enough. May I suggest you either (1) drop the 'outlier' points and extend the whiskers to the 2.5th and 97.5th percentiles? Or (2) consider a distributional plot like a violin plot. Also if this is for the appendix, could you make the fonts for the axis labels bigger and make the plot longer to accommodate? Not everyone is young and able to read tiny fonts. (Same for the appendix figure 5)

Response: To summarize the simulation results, we have chosen to drop simulation points from the plot that lie outside the range encompassed by the 0.025 to 0.975 quantiles such that the whiskers in the boxplots indicate the 95% simulation interval. The edges of the box cover the 0.25 to 0.75 quantiles, and the heavy vertical line inside the box is the median value. We feel that these changes more clearly characterize the sampled distribution of simulated bias-corrected cumulative infections. We have also increased the font size on appendix Figures 2 and 5 as suggested by the reviewer.

Reviewer #2 (Remarks to the Author):

The authors have made a number of improvements to the earlier version of the paper in response to my comments and those of other reviewers. The description of the methods is generally improved and the paper is strengthened the addition of at least some sensitivity analyses.

14. One or two concerns remain. The results of the paper remain quite unsurprising perhaps – mainly that it is not possible with any certainty to ascertain the total number of covid-19 cases and, consequently the degree of herd immunity.

Response: We thank the reviewer for this comment. While we agree that the findings are not necessarily surprising, in our opinion, this paper makes an important contribution by illustrating

the utility of the probabilistic bias analysis approach. We believe this approach is highly applicable to future COVID-19 studies, particularly since testing capacity remains very low in the U.S. and numerous other countries, even seven months into the pandemic. In addition, by describing our method in this manuscript and making our code fully available, we hope to facilitate the adoption of this approach in future COVID-19 and non-COVID-19 studies. With regard to herd immunity, while the simulation interval on our estimates is wide, even if we were to use the upper bound of our estimates (14,856,756 by mid-April), the proportion infected in the US was <5% — far lower than a realistic threshold to achieve natural herd immunity.

Some more specific comments are given below:

15. P. 8 The parameters alpha and beta are introduced before they are defined and the current definition remains unclear. It would be useful to define these in the text before first use. One has to infer the definition of these are the ratios appearing between the upper and lower bounds α_0 , α_1 , and β_0 and β_1 .

Response: We have substantially restructured the beginning of the Methods section in response to this comment as well as comment #2. We defined the parameters at the location where they first appear to help clarify the relationship between these values, the empirical data from states, and the estimated quantity. These definitions now appear in the following part of the methods section on page 7-8.

“*Sampling prior distributions*”

We defined prior distributions for seven parameters: $P(S_1|\text{tested})$, $P(S_1|\text{untested})$, $P(S_0|\text{test +})$, α , β , SARS-CoV-2 test sensitivity and specificity based on available evidence (Table 1, Appendix Figure 1, Appendix Table 2). We define S_1 to be an indicator variable of moderate to severe COVID-19 symptoms, and S_0 to be an indicator variable of minimal or no COVID-19 symptoms. $P(S_1|\text{tested})$ is the probability of having moderate to severe symptoms among tested individuals, and $P(S_1|\text{untested})$ is the analogous probability among untested individuals. $P(S_0|\text{test +})$ is the probability of having mild or no symptoms among individuals who tested positive. We defined α and β as random variables describing the ratio of $P(\text{test +} | S_1, \text{untested})$ and $P(\text{test +} | S_0, \text{untested})$ to the empirical state level estimate $P(\text{test+} | \text{tested})$.

For each state, we calculated the empirical estimate $P(\text{test+} | \text{tested})$ as the cumulative number of cases divided by the cumulative number of tests in each state from February 28, 2020 to April 18, 2020. Therefore $P(\text{test+} | \text{tested})$ is a point estimate of the average probability over this interval. We chose not to vary this quantity by date because the low testing rates per state resulted in unstable estimates of $P(\text{test+} | \text{tested})$, particularly when less than 1% of the population was tested, which was the case in most states until early-mid April (Appendix Figure 3). Furthermore, when restricting to dates when at least 0.6% of the population was tested in each state, $P(\text{test+} | \text{tested})$ was relatively stable within each state over time, suggesting that at least over the period considered, this quantity did not change significantly over time (Appendix Figure 4).

We sampled 10^4 values from the distributions of $P(S_1|\text{tested})$, $P(S_1|\text{untested})$, α , β , test sensitivity and test specificity. Using the state-level empirical estimates of $P(\text{test} + | \text{tested})$ and sampled values of α , β , we sample $P(\text{test} + | S_1, \text{untested})$ and $P(\text{test} + | S_0, \text{untested})$, the probability an untested individual exhibiting moderate to severe symptoms (likewise for none to mild) would test positive on their initial test. For untested individuals who had moderate to severe symptoms, we assumed that the probability they tested positive was 80% to 99% the empirical probability of testing positive among tested individuals in a state, $P(\text{test} +|\text{tested})$. Accordingly, α was defined such that $\alpha_0 < P(\text{test} + |S_1, \text{untested})/P(\text{test} +|\text{tested}) < \alpha_1$, with truncated support between $\alpha_0 = 80\%$ and $\alpha_1 = 99\%$. For untested individuals who had mild or no symptoms, we assumed that the probability they tested positive was 0.2% to 32% of the empirical probability. Likewise, β was defined such that $\beta_0 < P(\text{test} + |S_0, \text{untested})/ P(\text{test} +|\text{tested}) < \beta_1$, with truncated support on $\beta_0 = 0.2\%$ and $\beta_1 = 32\%$.”

16. Why should the test sensitivity and specificity be uncorrelated across states (p, 8)?

This seems like a very implausible assumption. The prior you place on these parameters represents your belief about the accuracy of the test. Undoubtedly positive correlations between sensitivity across states would lead to a even greater uncertainty in the total number of cases at the national level? More discussion of this point, and justification of the independence assumption should be given.

Response: We have added a paragraph to the Methods section with the subheading “Independence of parameters between states” (page 8) to discuss the reasons behind this assumption. While we acknowledge that in principle it is possible that this assumption is violated, we expect that the extent of correlation between states is likely to be minimal for the following reasons:

- Clinical laboratory practice is regulated at the federal level, so laboratory practices and the types of samples collected (e.g., nasopharyngeal swabs vs. throat swabs) — both of which could affect diagnostic accuracy of RT-PCR — should not be subject to strong between state correlations.
- Sensitivity may be lower for tests of individuals with lower SARS-CoV-2 viral loads, and viral loads may be correlated with symptom presentation. If there are between-state correlations in testing based on symptom presentation, this could in turn result in between-state correlation in diagnostic accuracy. However, there is some evidence that asymptomatic and symptomatic individuals have similar viral loads (Long et al., 2020, Zheng et al., 2020). Thus, even if testing protocols for asymptomatic vs. symptomatic individuals were correlated between states, based on this evidence we would not expect this specific factor to contribute strongly to correlations in diagnostic accuracy between states.
- Unfortunately, we are not aware of any datasets that could be used to assess whether there is spatial correlation between state-level sensitivity/specificity nor the specific correlation structure, and we present results with the caveat that accounting for this

correlation structure would produce wider uncertainty intervals. We also wish to note that the prior distributions we placed upon sensitivity and specificity represent our beliefs about the performance of tests averaged across states. Specifying a correlation structure across states would not change the marginal distributions of sensitivity or specificity.

We have added the following text in the Methods section with the subheading “Independence of parameters between states” (page 8) to more clearly discuss this assumption and the lack of data to support its parameterization.

“We assumed that the parameters, $P(S_1|tested)$, $P(S_1|untested)$, $P(S_0|test +)$, α , β , test sensitivity and test specificity were independent and identically distributed across states, whereas $P(test+ | tested)$ and consequently $P(test + | S_0, unttested)$, $P(test + | S_1, unttested)$ varied between states based on empirical data. In principle, it is possible that some or all of these parameters were correlated between states. Between-state correlation in testing protocols for asymptomatic vs. symptomatic individuals may have existed; however, to our knowledge there is no evidence to inform assumptions about correlation between states during the study period. State-level correlation in testing probabilities (e.g., the probability that a mild or asymptomatic individual was tested) could affect test sensitivity as lower viral loads are less likely to be detected.³⁹ However, there is evidence that asymptomatic and symptomatic individuals have similar viral loads, and as such, we would not expect sensitivity to vary by state during the study period.^{40,41} During the early epidemic period, there was also no evidence that laboratory practices or the type of sample collected for RT-PCR (e.g., nasopharyngeal vs. throat swabs), both of which could affect diagnostic accuracy, varied substantially by state. To our knowledge, there is no available data for the study period that could be used to reasonably assess between-state correlation in parameters to estimate an appropriate correlation matrix. For these reasons, we assumed parameters were uncorrelated between states, with the caveat that if this assumption is incorrect, simulation intervals for estimated cumulative infection burden will be too narrow.”

17. Table 2 – are you not conditioning on being untested also here i.e. $P(test + |S_0, unttested)$? If so, there is an inconsistency between what is stated in the text and the table which should be corrected.

Response: We appreciate the careful reading of the table, and we have corrected the column labels to correctly indicate that we are indeed conditioning on being untested as well.